# A Machine Learning-Based Risk Prediction Model for Post-Traumatic Stress Disorder during the COVID-19 Pandemic

**DOI:** 10.3390/medicina58121704

**Published:** 2022-11-22

**Authors:** Yang Liu, Ya-Nan Xie, Wen-Gang Li, Xin He, Hong-Gu He, Long-Biao Chen, Qu Shen

**Affiliations:** 1Department of Nursing, School of Medicine, Xiamen University, Xiamen 361102, China; 2Department of Clinical Medicine, School of Medicine, Xiamen University, Xiamen 361102, China; 3Department of Computer Science, School of Informatics, Xiamen University, Xiamen 361005, China; 4Alice Lee Centre for Nursing Studies, Yong Loo Lin School of Medicine, National University of Singapore, Singapore 117597, Singapore

**Keywords:** COVID-19, machine learning, stress disorders, post-traumatic, mental health, risk prediction model

## Abstract

*Background and Objectives*: The COVID-19 pandemic has caused global public panic, leading to severe mental illnesses, such as post-traumatic stress disorder (PTSD). This study aimed to establish a risk prediction model of PTSD based on a machine learning algorithm to provide a basis for the extensive assessment and prediction of the PTSD risk status in adults during a pandemic. *Materials and Methods*: Model indexes were screened based on the cognitive–phenomenological–transactional (CPT) theoretical model. During the study period (1 March to 15 March 2020), 2067 Chinese residents were recruited using Research Electronic Data Capture (REDCap). Socio-demographic characteristics, PTSD, depression, anxiety, social support, general self-efficacy, coping style, and other indicators were collected in order to establish a neural network model to predict and evaluate the risk of PTSD. *Results*: The research findings showed that 368 of the 2067 participants (17.8%) developed PTSD. The model correctly predicted 90.0% (262) of the outcomes. Receiver operating characteristic (ROC) curves and their associated area under the ROC curve (AUC) values suggested that the prediction model possessed an accurate discrimination ability. In addition, depression, anxiety, age, coping style, whether the participants had seen a doctor during the COVID-19 quarantine period, and self-efficacy were important indexes. *Conclusions*: The high prediction accuracy of the model, constructed based on a machine learning algorithm, indicates its applicability in screening the public mental health status during the COVID-19 pandemic quickly and effectively. This model could also predict and identify high-risk groups early to prevent the worsening of PTSD symptoms.

## 1. Introduction

The 2019 novel coronavirus disease (COVID-19) pandemic has globally affected people’s health since its outbreak, especially after the World Health Organization [1] declared it a public health emergency of international concern on 30 January 2020. The COVID-19 pandemic has strained health systems worldwide with a burden of severe illness and deaths [2]; this has frequently led to clusters of infections in homes and hospitals. By 9 September 2022, the total global numbers of confirmed cases and deaths were 603,711,760 and 6,484,136, respectively [3].

Owing to the sudden outbreak of COVID-19, its highly contagious nature, and the social isolation policy adopted by many countries worldwide, the COVID-19 pandemic has had a profound impact on health and well-being worldwide. It has also generated immense psychological pressure on the global population. The international population has exhibited several symptoms, including insomnia, loneliness, frustration, helplessness, anxiety and depression [4,5,6,7], post-traumatic stress disorder (PTSD), and other serious mental diseases [8,9]. A previous study examining the impact of severe acute respiratory syndrome (SARS) on long-term mental illness found that PTSD was the most common long-term mental disorder, with the cumulative incidence reaching 47.8% thirty months after the SARS outbreak [10]. Diagnostic PTSD prevalence estimates range from 7% to 33.2% across diverse data from different countries, such as China [11], the United Kingdom [12], India [13], and the U.S. [14]. Therefore, during the COVID-19 pandemic, there is an imperative need to strengthen the capacity of mental health personnel to monitor and intervene in possible PTSD among the general population.

Several studies have shown that the COVID-19 pandemic is a traumatic event that has caused people to develop PTSD [8,15]. Additionally, some relevant studies have reported that factors such as gender, exposure history of the outbreak site, high-risk groups for infection, isolation, and negative emotions could be predictive factors of PTSD during the COVID-19 pandemic [11,16]. Other factors, such as fear of infection, financial problems, the uncertainty of future life, and personal life limitations, might also influence the symptoms of PTSD [17].

On 27 January 2020, the Chinese Health Commissioner of the Disease Prevention and Control Department published the first policy guidelines for psychological crisis intervention during the COVID-19 pandemic [18]. Interventions should be tailored to different populations; for example, while mental health education would be suitable for the general population, psychological crisis interventions would be necessary for high-risk groups. Therefore, it is necessary to establish relevant psychological risk prediction models during the COVID-19 pandemic to screen which groups need more support. There is a lack of studies that have used machine learning algorithms to construct PTSD risk prediction models for the quick preliminary screening of psychological conditions among the global public population. Therefore, this study aimed to construct a risk prediction model of PTSD based on machine learning algorithms for psychological monitoring during the COVID-19 pandemic. Our core research strategy is to design questionnaires to investigate the exposed population by summarizing the factors that influence the onset of PTSD. Using machine learning models, algorithms were designed to automatically analyze big data to screen out important factors that can predict PTSD risk exposure and establish relevant psychological screening models.

## 2. Materials and Methods

### 2.1. Design, Participants, Data Collection, and Ethical Considerations

This was a descriptive, cross-sectional correlational study. Research Electronic Data Capture (REDCap, Vanderbilt University, Nashville, TN, USA) was used to collect data in China through Chinese WeChat, QQ, and microblog social media from 1 March to 15 March 2020. Residents (≥18 years old) of mainland China were recruited via a convenient sampling method during the COVID-19 pandemic. This study was approved by the Medical Ethics Committee of the School of Medicine, Xiamen University (No. XDYX2020005). The participants agreed to participate in the online questionnaire and click the confirmation option of the questionnaire. Clicking the “Agree” button was deemed to indicate consent to participate in this study.

### 2.2. Outcomes and Measurement

The demographics collected comprised gender, age, ethnicity, political status, educational background, marital status, employment status, whether the respondent was medical staff, family members, and personal monthly income. Some questions were also asked to evaluate COVID-19-related situations that participants had experienced, including exposure to Wuhan, traveling by public transport, experience of seeing a doctor, isolation due to COVID-19 restrictions, and experience of having confirmed COVID-19 cases around. These situations were measured using self-reported questions.

PTSD symptoms were measured using the PTSD Checklist—Civilian Version (PCL-C) [19]. The PCL-C comprises seventeen 5-point Likert items, scored from 1 (not at all) to 5 (extremely). Higher scores indicate more severe symptoms, and a score of 38 was considered to be a cut-off point for PTSD screening [20,21]. The PCL-C has demonstrated good reliability and validity among Chinese samples [9,22]. The Cronbach’s α value was found to be 0.86 and the test–retest reliability was 0.88 [23].

Depression was measured via the Patient Health Questionnaire (PHQ-9) [24], which consists of nine items scored from 0 (never) to 3 (almost daily). Higher total scores indicate more severe symptoms, with scores greater than 5 indicating that an individual is likely to be depressed [25]. The PHQ-9 has been widely used among the Chinese population and has demonstrated good reliability and validity [9,26,27]. The Cronbach’s α value was found to be 0.89 [27].

Anxiety was measured using Zung’s Self-rating Anxiety Scale (SAS) [28], which consists of 20 items scored from 1 (a little of the time) to 4 (most of the time). Higher scores indicate more severe symptoms. The scale has been widely used among the Chinese population. The SAS has demonstrated good reliability and validity [29] and has been proven to have good internal consistency, with a Cronbach’s α value of 0.86 [30].

Social support was measured via the Perceived Social Support Scale (PSSS) [31], which comprises 20 items scored from 1 (very strongly disagree) to 7 (very strongly agree). Higher scores indicate higher levels of perceived social support. The Cronbach’s α value of the PSSS was found to be 0.88 and the test–retest reliability was 0.85 [32,33].

Participants’ self-efficacy was measured using the General Self-Efficacy Scale (GSES) [34,35], which comprises 10 items scored from 1 (not at all true) to 4 (exactly true). A higher score indicates a higher level of self-efficacy. The scale has been widely used among the Chinese population and has exhibited good reliability and validity [36]. The internal consistencies as measured by Cronbach’s α were found to be 0.84, 0.81, and 0.91, respectively [34].

Coping style was measured using the Self-reported Coping Style (SCS) scale. The SCS was adapted from the Ways of Coping Questionnaire (WCQ) [37] for the Chinese population by Xie in 1998 [38], with acceptable psychometrics (Cronbach’s α coefficient 0.89, comprising two factors). It was used to evaluate the individual’s conscious and purposeful flexible adjustment behavior to the change in the real environment. The scale consists of two dimensions and 20 items scored from 0 (take no action) to 3 (take active action), with a total score of 60, among which 0–12 items constitute the positive coping dimension and 13–20 items constitute the negative coping dimension. Higher scores indicate better coping styles.

### 2.3. Model Parameters and Indexes

During the process of model construction, we mainly determined the parameters and indexes of the risk warning model of PTSD. The model primarily comprised 20 indexes, including general demographic data, depression, anxiety, social support, general self-efficacy, and coping style, among others. The specific parameters and indexes were set as follows.

The neural network model system was set up to include one input layer, two hidden layers, and one output layer, with the numbers of neurons in each layer being 62, 12, 9, and 2, respectively. The activation function of the hidden layer was the sigmoid function, the output layer was the softmax function, and the error function was cross-entropy. The remaining parameters were treated with the default values of the architecture. The partition variable was set to partition the data set, dividing it into training samples (1157, 56.1%), inspection samples (613, 29.7%), and test samples (291, 14.1%). The multi-layer perceptron (MLP) method was used to analyze, learn, and train the data. Finally, the model was optimized by automatically changing the weight values of nodes in each layer through the gradual learning of data. Schematic illustrations of the neural network model are presented in Figure 1 and Table 1.

### 2.4. Statistical Analysis

Statistical analysis was performed using SPSS Statistics version 26 (IBM Corporation, Armonk, NY, USA). Descriptive statistics of the mean and standard deviation were used to report the socio-demographic characteristics, PTSD, depression, anxiety, social support, self-efficacy, and coping style; additionally, the categorical variables and dichotomous variables were expressed in frequencies and percentages (%). Following this, we also used machine learning methods, including a neural network to build our model to predict the risk of PTSD within the general population during the COVID-19 pandemic, and tested the accuracy of the model. Finally, we estimated the contribution of each index to the model.

## 3. Results

We received a total of 2067 valid questionnaires. The majority of the participants were female (*n* = 1598; 77.3%), were <25 years old (*n* = 1162; 56.2%), and had a junior college degree (*n* = 1903; 92.1%). More than one-third of them (*n* = 801; 38.8%) worked in hospitals. Of the 2067 participants, 1085 (52.5%) engaged in negative coping behaviors, 441 (21.3%) had poor social support, and 788 (38.2%) had low self-efficacy. The psychological status and behavior of the target population were noteworthy. Of all participants, 378 (18.3%) exhibited some level of anxiety, 699 (33.8%) had depressive tendencies, and 368 (17.8%) had symptoms of PTSD, indicating varying degrees of psychological reactions to the COVID-19 pandemic in China (Table 2). The employed scales had acceptable reliability in this study, with Cronbach’s α coefficients of 0.96, 0.91, 0.88, 0.96, 0.94, and 0.90 for PCL-C, PHQ-9, SAS, PSSS, GSES, and SCS, respectively.

As shown in Table 3, the artificial neural network (ANN) model correctly predicted 262 (90.0%) outcomes. In the external test set, 43 (70.5%) of the positive samples were correctly identified and classified compared with the correct identification and classification of 219 (95.2%) negative samples.

Apart from the forecast results, we also tested the performance of the neural network models. Receiver operating characteristic (ROC) curves plotted the sensitivity (percentage of positive tests correctly predicted as positive by the model) against one minus the specificity (percentage of negative tests correctly predicted as negative by the model). The ROC curve for the predictive power of the model is presented in Figure 2. The closer the ROC curve is to the upper-left corner of the graph, the higher the accuracy of the test. The area under the ROC curve (AUC) value was 0.893. This suggests that the prediction model had a good level of discriminative ability (Figure 2).

Furthermore, we tested the contribution of each factor (Figure 3). Depression and anxiety were the most important indicators in the model, with standardized importance values of 100% and 62.4%, respectively. In addition, coping style, social support, age, self-efficacy, and exposure to the city of Wuhan were also key indicators, the standardized importance values being 42.2%, 35.6%, 35.2%, 34.6%, and 33.0%, respectively. In contrast, whether participants were professional medical staff, as well as personal monthly income, gender, employment, isolation, education, and marital status contributed little to the model, with standardized importance values below 10.0%.

The contributions of the depression and anxiety predictors to the model were 100.0% and 62.4%, respectively. Additionally, negative emotions were found to represent the most important risk predictor of PTSD. It was revealed that prevalence rates of 28.2% for PTSD and 14.1% for depression were observed during the COVID-19 pandemic. A significant (*p* < 0.05) association was reported between depression and PTSD (Pearson correlation coefficient of 0.649), revealing that symptoms of PTSD and depression were common during home quarantine [39]. Coping style predictors contributed 42.2% to the model, social support contributed 35.6%, and general self-efficacy predictors contributed 34.6%. Thus, it can be concluded that individual coping styles, social support, and general self-efficacy had strong abilities to predict the risk of PTSD symptoms. The prediction results showed that age (35.2%) was an essential predictor of demographic characteristics in the predictive model. Personal experience predictors associated with COVID-19 outbreaks contributed to the model as follows: exposure to the city of Wuhan (33.0%), travel by public transport (10.9%), the experience of seeing and consulting a doctor (10.9%), and whether there were COVID-19 patients in the vicinity of the individual (10.9%). The results revealed that patients with COVID-19 or those with a history of living in Hubei Province were more likely to suffer from PTSD. In addition, individuals who had undergone the experience of medical treatment, had interpersonal relationships with COVID-19 patients, and have a history of living in Hubei Province were more likely to come into contact with infected patients or suspected COVID-19 patients. In addition, family members contributed 14.6% to the model: the greater the number of family members, the more severe the PTSD symptoms. Personal monthly income contributed to the model at 8.0%. The results showed that the higher the individual’s average monthly income, the more likely they were to experience anxiety and depression, and develop PTSD symptoms.

## 4. Discussion

This study was the first to use a machine learning algorithm to construct a risk prediction model of PTSD for a general population during the COVID-19 pandemic. In our study, 378 (18.3%) and 699 (33.8%) participants reported various anxiety levels and depressive tendencies, respectively; these rates were higher than those reported in a previous study, where 8.3% and 14.6% of 1593 participants reported anxiety and depression, respectively [40]. Moreover, our study found that 368 (17.8%) participants had PTSD symptoms, which was also higher than the rates reported in previous studies of 15.8% [41], 7.6% [42], and 7% [11]. The government must focus more on providing economic and medical support to improve the mental health of the general population. Therefore, we screened the appropriate model prediction indexes and established a risk prediction model of PTSD based on a machine learning algorithm to help predict psychological conditions among the general population. We also tested the performance of the model. The insights gained from this study can help to evaluate the mental health of the population, especially those living in areas that have been severely affected by the pandemic.

The predictors we obtained were valid, which has been validated by many other studies. It has been found that depression and anxiety symptoms may be precursory manifestations of PTSD; additionally, depression, anxiety disorder, and PTSD are highly comorbid [43,44]. However, PTSD symptoms appear days or months after the trauma and can last for years. Therefore, it is necessary to assess the risk of PTSD and intervene in a population experiencing negative emotions, such as symptoms of depression or anxiety, before the symptoms of PTSD become apparent or worsen.

Relevant studies have found that coping style moderated the stress–emotional distress relationship, i.e., individuals who mainly adopted positive coping strategies suffered fewer symptoms of depression, compulsion–anxiety, and neurasthenia under stress, while negative coping strategies aggravated emotional distress [45]. Therefore, the public should be guided to scientifically understand the pandemic situation, rationally identify information, reasonably vent negative emotions, actively seek psychosocial support, avoid negative behaviors, and maintain a normal life as much as possible to ensure their physical and emotional safety. A high level of social and subjective support, the network connecting one to the outside world, and the degree of subjective feeling and evaluation cushion the impact of the pandemic, to some extent, reduce the negative effects of the COVID-19 pandemic, and are advantageous in maintaining an individual’s positive state of mind. In contrast, insufficient social support can make individuals feel hopeless, further leading to deteriorating psychological conditions and negative emotions. The level of self-efficacy reflects an individual’s perception of their ability to cope with the pandemic successfully and affects the individual’s thought process and emotional response during a pandemic. The higher the level of self-efficacy and the higher the individual’s confidence in coping with the pandemic, the more positive their attitude and the lower their likelihood of developing PTSD symptoms.

The results revealed that individuals aged 26 to 35 had the most severe PTSD symptoms; this may be because young and middle-aged people have been exposed to more COVID-19-related information and consultation. As they have more COVID-19 prevention and control knowledge, they are more likely to be nervous and cautious, and experience more anxiety and tension. Ahmed et al. (2020) [46] and Huang and Zhao (2020) [47] also demonstrated that individuals under 40 years of age exhibited more adverse psychological symptoms during the COVID-19 pandemic than the older population.

People involved in COVID-19 outbreaks are often more worried about their safety and that of their families and friends than other groups; thus, they are more likely to experience anxiety and depression, and develop PTSD symptoms. Therefore, during the pandemic prevention and control period, special attention should be paid to people who visit hospitals, travel where there are confirmed COVID-19 cases, or have a history of living in and traveling in and around Hubei Province. To date, China has issued several psychological assistance policies, mainly psychological hotline counseling. Psychological hotline counseling can alleviate psychological crises for some people, but many people may have undetected psychological stress symptoms. Therefore, interventions should be implemented, and psychological assistance should be provided as early as possible, with extensive preliminary psychological screening to identify and treat high-risk groups

As large families are more exposed, we need to raise awareness regarding the prevention and control of family clusters. Family members should not gather or share personal articles for daily use and should eat separately, wash hands frequently, ventilate frequently, maintain a balanced diet, exercise appropriately, work and rest regularly, and improve their physical fitness and immunity.

High levels of PTSD were recorded among the general Chinese population, and they were strongly associated with increased concern regarding COVID-19. The factors affecting the model may facilitate the detection of factors that may be modifiable to help enable mental illness prevention and recovery. Therefore, the establishment of an early and sound psychological warning and intervention system based on artificial intelligence (AI) technology, as well as the development of relevant psychological intervention-supporting procedures based on the big data health model after universal screening, will be more efficient and feasible in solving psychological problems among the general population during a pandemic.

In the future, we hope to develop a simple, feasible, and robust support policy to meet the mental health challenges arising from the COVID-19 pandemic, taking into account definite local and cultural characteristics and based on the risk prediction model of PTSD built in this study, along with the continuous improvement of follow-up psychological intervention policies. The ultimate goal is to provide practical psychological assistance and technical support to mitigate the damaging effects of national disasters and emergencies.

### Limitations

Our study has several limitations. First, the depression, anxiety, self-efficacy, coping styles, social support, and PTSD scores were measured through self-reporting scales. As with all self-reported measurements, common methodological variance, social desirability biases, and response distortion due to ego-related defensive tendencies cannot be ignored. Second, our participants mainly had homogenous demographic characteristics concerning gender and age. Though there were a few positive PTSD samples, the learning and recognition ability of the model for those who were PTSD-positive needed improvement. Third, the target population was Chinese adults. The social and cultural context of China should be considered when interpreting the results. When the target population changes, the risk factors may differ; therefore, the prediction indicators should be adjusted and re-considered reasonably and accordingly when applying the results of this study to other populations. Finally, in the selection of indicators, this study partly drew on the opinions of domestic and foreign scholars and made adjustments according to the actual situation. This may have resulted in certain subjective opinions regarding the model indicators.

## 5. Conclusions

The COVID-19 pandemic has impacted the psychological state of the general population in China. The risk prediction model established in this study can be used to identify high-risk groups for future public health emergencies. Future mental health-based psychological interventions are needed to improve the mental health of those in need.

### Relevance for Clinical Practice

The machine learning algorithm could predict mental health problems during the COVID-19 pandemic with high sensitivity and could potentially be used as a screening tool in clinical practice. In addition, it offers a novel approach to help agencies make timely predictions and interventions for future public health emergencies.

## Figures and Tables

**Figure 1 medicina-58-01704-f001:**
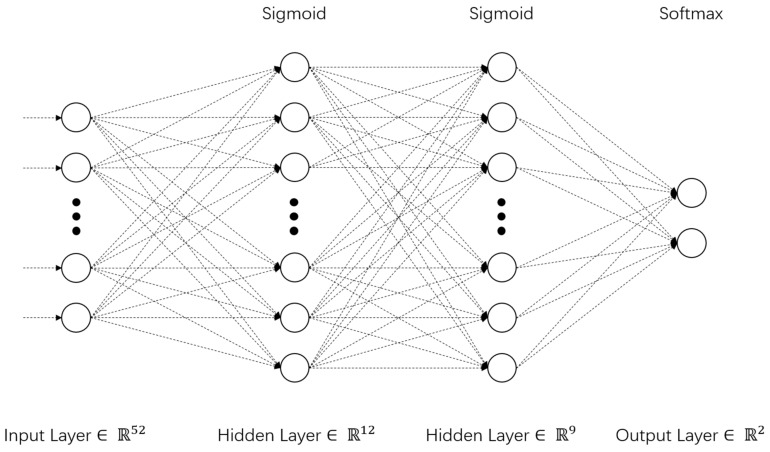
Model parameters and indexes.

**Figure 2 medicina-58-01704-f002:**
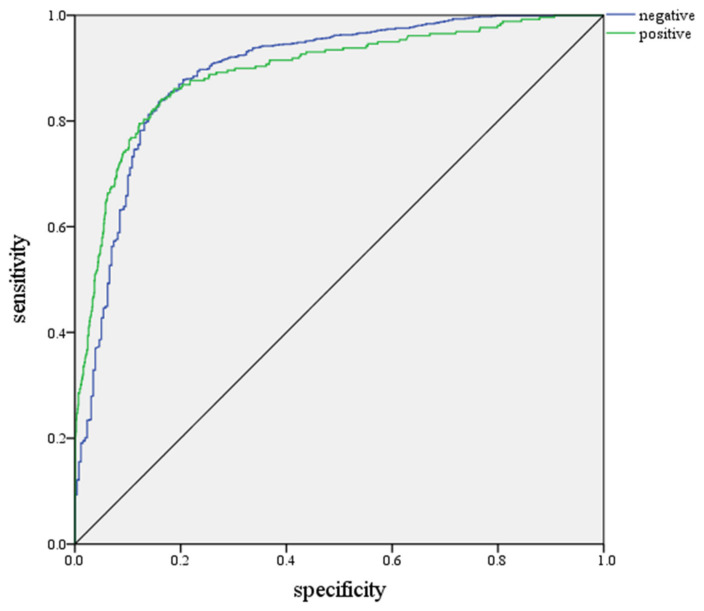
ROC Curve. ROC = 0.893.

**Figure 3 medicina-58-01704-f003:**
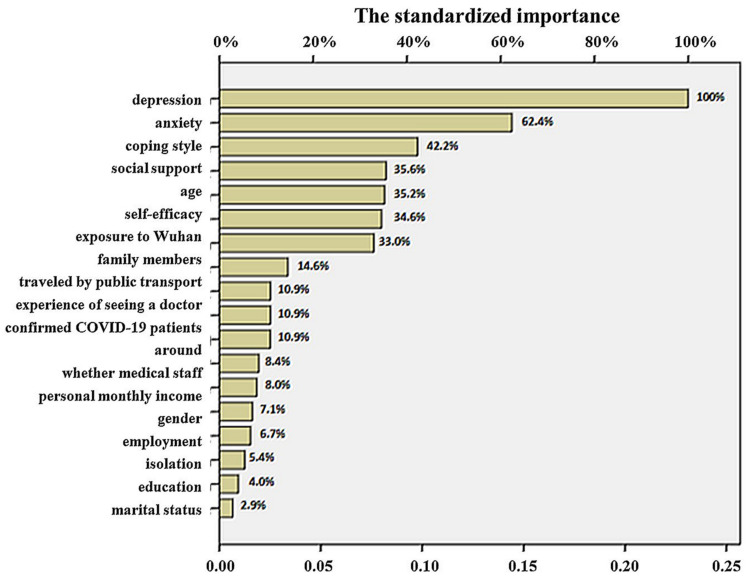
Importance of indexes.

**Table 1 medicina-58-01704-t001:** Model parameters and indexes (*n* = 2067).

Input Layer	Indexes	1	Gender	
		2	Age (Years)	
		3	Ethnicity	
		4	Political status	
		5	Educational background	
		6	Marital status	
		7	Family members	
		8	Personal monthly income	
		9	Employment	
		10	Whether medical staff	
		11	Exposure to Wuhan	
		12	Traveled by public transport	
		13	Experience of seeing a doctor	
		14	Isolation	
		15	Confirmed COVID-19 patients around	
		16	Depression	
		17	Anxiety	
		18	Social support	
		19	Self-efficacy	
		20	Coping style	
	Units ^a^			52
Hidden Layer	Number of Hidden Layers			2
	Units ^a^ of Hidden Layer 1			12
	Units ^a^ of Hidden Layer 2			9
	Activation Function		Sigmoid	
Output Layer	Dependent Variable	1	PTSD	
	Units ^a^			2
	Activation Function		Softmax	
	Error Function		Cross Entropy	

^a^ Eliminate bias unit.

**Table 2 medicina-58-01704-t002:** Descriptions of all indexes (*n* = 2067).

Variable	Category	*n* (%)	PTSD (PCL-C)	*p* Value
Gender	Male	469 (22.7)	32.2 (15.1)	<0.001
	Female	1598 (77.3)	27.1 (10.7)	
Age (Years)	≤25	1162 (56.2)	27.9 (12.2)	0.123
	26–35	643 (31.1)	29.2 (12.3)	
	36–45	177 (8.6)	27.6 (10.6)	
	>45	85 (4.1)	27.8 (10.3)	
Education	Senior high school and below	164 (7.9)	31.7 (15.8)	<0.001
	Junior college	1069 (51.7)	26.9 (10.6)	
	Bachelor‘s and above	834 (40.3)	29.3 (12.6)	
Marital status	Unmarried	1458 (70.5)	28.1 (12.1)	0.291
	Married	609 (29.5)	28.7 (11.8)	
Family members	≤2	153 (7.4)	28.4 (12.6)	0.024
	3–4	1407 (68.0)	27.8 (11.6)	
	≥5	507 (24.6)	29.5 (13.0)	
Personal monthly income	≤¥3500	495 (23.9)	26.1 (10.3)	<0.001
	¥3501–¥5000	490 (23.7)	27.9 (11.7)	
	¥5001–¥8000	502 (24.3)	28.9 (12.9)	
	¥8001–¥12,500	345 (16.7)	30.5 (13.2)	
	≥¥12,501	235 (11.4)	29.0 (11.8)	
Medical staff or not	Yes	801 (38.8)	27.5 (10.6)	0.018
	No	1266 (61.2)	28.7 (12.9)	
Exposure to Wuhan	Yes	62 (3.0)	44.9 (17.6)	<0.001
	No	2005 (97.0)	27.7 (11.5)	
Traveled by public transport	Yes	407 (19.7)	30.9 (13.4)	<0.001
	No	1660 (80.3)	27.6 (11.6)	
Experience of seeing a doctor	Yes	193 (9.3)	36.4 (15.3)	<0.001
	No	1873 (90.6)	27.4(11.3)	
Isolation	Yes	161 (7.8)	34.0 (15.3)	<0.001
	No	1906 (92.2)	27.8 (11.6)	
Confirmed COVID-19 patients around	Yes	84 (4.1)	43.6 (16.9)	<0.001
	No	1983 (95.9)	27.6 (11.3)	
PTSD (PCL-C)	Negative	1699 (82.2)	-	-
	Positive	368 (17.8)	-	
Depression (PHQ-9)	Healthy	1368 (66.2)	22.8 (6.6)	<0.001
	Depressed	699 (33.8)	39.0 (13.0)	
Anxiety (SAS)	Normal	1688 (81.7)	24.5 (7.9)	<0.001
	Mild	245 (11.9)	40.2 (10.9)	
	Moderate	90 (4.4)	49.7 (11.7)	
	Severe	43 (2.1)	61.6 (10.9)	
Social support (PSSS)	Severe	66 (3.2)	33.8 (15.4)	<0.001
	Mild	375 (18.1)	32.0 (12.6)	
	Normal	1626 (78.7)	25.4 (10.4)	
Self-efficacy (GSES)	Lower	788 (38.1)	30.6 (12.0)	<0.001
	Average	632 (30.6)	28.8 (11.9)	
	Higher	643 (31.1)	24.4 (11.5)	
Coping style (SCS)	Negative	1085 (52.5)	31.9 (13.7)	<0.001
	Positive	982 (47.5)	24.3 (8.2)	

**Table 3 medicina-58-01704-t003:** Model prediction results (*n* = 2067).

Observed Value	Predicted Value		
	Negative	Positive	Correctly Predicted Percentage
Negative	219	11	95.2%
Positive	18	43	70.5%
Overall percentage	81.4%	18.6%	90.0%

## Data Availability

The data are contained within the article. The datasets are available from the corresponding authors on reasonable request.

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
