# Peer review of "A Machine Learning-Based Risk Prediction Model for Post-Traumatic Stress Disorder during the COVID-19 Pandemic"

_medicina, 2022, doi:10.3390/medicina58121704_

Round 1

Reviewer 1 Report

The manuscript prepared by Liu et al, entitled "A machine learning-based risk prediction model for post-traumatic stress disorder during the COVID-19 pandemic" is particularly relevant in the current scenario of the emergence of viral infectious diseases with strong potential to constitute global threats to mental health. I agree that it is indeed time for equal attention to COVID-19 pandemic and mental health by global healthcare providers, researchers, and local governments.

Author Response

It is of great pleasure for us to receive your comments, and thank you for your appreciation of the paper. We would like to thank you for your constructive comments on our manuscript.

We have modified our manuscript accordingly, and responded all comments from you below.

Point 1: English language and style: English language and style are fine/minor spell check required

Response 1: After the revision, we invited a native English-speaking colleague from Canada to improve the quality of English in the manuscript. There were a number of sentences adjusted across the text. We believe that the language now is acceptable for the review process.

Point 2: The manuscript prepared by Liu et al, entitled "A machine learning-based risk prediction model for post-traumatic stress disorder during the COVID-19 pandemic" is particularly relevant in the current scenario of the emergence of viral infectious diseases with strong potential to constitute global threats to mental health.

Response 2: We thank the reviewer for this comment. We have modified the introduction section to make it more clear on providing a background of this research area and the rationale of our study.

Point 3: I agree that it is indeed time for equal attention to COVID-19 pandemic and mental health by global healthcare providers, researchers, and local governments.

Response 3: We thank the reviewer for this comment. Your precious opinion will greatly encourage us in our future work.

Thank you again for your time spending on our manuscript. We sincerely look forward to your further opinions, and hope to have the pleasure of seeing our publication! 

With best regards,
Qu Shen, On behalf of all authors
School of Medicine, Xiamen University
10-Nov-2022

Reviewer 2 Report

The topic of the manuscript “A machine learning-based risk prediction model for post-traumatic stress disorder during the COVID-19 pandemic” is interesting. However, some changes should be done before publication as there are some confusions and shortcomings that detract from the quality of the manuscript.

-The Introduction section should be revised, summarizing the relevant arguments and the past evidence to give the reader a firm sense of what was done and why. In addition, the research strategy used should be explained in greater depth.

-In the Introduction, on page 1, lines 39 and 40, it puts the following text: “The 2019 novel coronavirus disease (COVID-19) pandemic has globally affected people's health since the World Health Organization [1] declared it a public health emergency of international concern on January 30, 2020”. Said text should be revised since the affectation of people by the pandemic has not been since the World Health Organization declared it a public health emergency of international concern on January 30, 2020, but since the pandemic began.

-On Page 3, lines 89 to 92 describe the PTSD assessment instrument. On lines 90 and 91 it says “Higher scores indicate more severe symptoms, with a score ≥38 indicating PTSD symptoms”. This is confusing as all 17 PCL-C items are supposed to assess symptoms. Perhaps the authors mean that such a score is the cut-off point for the detection of posttraumatic stress disorder. This should be reviewed and corrected.

-All measurements used in the study should be described in the Measurement subsection (page 3). For example, in the text it is mentioned that coping style was evaluated, and results are presented, but in the Measurement section it is not mentioned which questionnaire or measure was used for the evaluation of coping styles.

-In “Table 1. Model parameters and indexes” (on page 4) there are variables that are not described in the Measurement section. And in said table Ethnicity appears 2 times but Education does not appear.

-The Discussion section should be thoroughly reviewed and instead of presenting so many results (many of which should be moved to the Results section), the discussion should focus on the evaluation, interpretation and implications of the results obtained. Also, on page 10, in lines 187 to 189, it says “Coping style moderated the stress-emotional distress relationship, i.e., individuals who mainly adopted positive coping strategies suffered fewer symptoms of depression, compulsion-anxiety, and neurasthenia under stress, while negative coping strategies aggravated emotional distress” but in the study presented there is no evidence that analyzes have been done to find moderating variables.

Author Response

It is of great pleasure for us to receive your comments, and thank you for your appreciation of the paper. We would like to thank you for your constructive comments on our manuscript. We have modified our manuscript accordingly, and responded all comments from you below.

Point 1: The topic of the manuscript “A machine learning-based risk prediction model for post-traumatic stress disorder during the COVID-19 pandemic” is interesting. However, some changes should be done before publication as there are some confusions and shortcomings that detract from the quality of the manuscript.

The Introduction section should be revised, summarizing the relevant arguments and the past evidence to give the reader a firm sense of what was done and why. In addition, the research strategy used should be explained in greater depth.

Response 1: In the introduction section, we have further described the research strategy in detail. Please view page 2 line 78-82 for the changes.

Point 2: In the Introduction, on page 1, lines 39 and 40, it puts the following text: “The 2019 novel coronavirus disease (COVID-19) pandemic has globally affected people's health since the World Health Organization [1] declared it a public health emergency of international concern on January 30, 2020”. Said text should be revised since the affectation of people by the pandemic has not been since the World Health Organization declared it a public health emergency of international concern on January 30, 2020, but since the pandemic began.

Response 2: We have modified this statement to “… since its outbreak, especially after the WHO’s declaration of the PHEIC”. Thanks for your carefully reviewing our text. Please view page 1 line 39-40 for the changes.

Point 3: On Page 3, lines 89 to 92 describe the PTSD assessment instrument. On lines 90 and 91 it says “Higher scores indicate more severe symptoms, with a score ≥38 indicating PTSD symptoms”. This is confusing as all 17 PCL-C items are supposed to assess symptoms. Perhaps the authors mean that such a score is the cut-off point for the detection of posttraumatic stress disorder. This should be reviewed and corrected.

Response 3: Yes, this is exactly what we want to say. A score of 38 was considered to be the cut-off point for PTSD screening. We have adjusted the statement accordingly. Please view page 3 line 92 for the changes.

Point 4: All measurements used in the study should be described in the Measurement subsection (page 3). For example, in the text it is mentioned that coping style was evaluated, and results are presented, but in the Measurement section it is not mentioned which questionnaire or measure was used for the evaluation of coping styles.

Response 4: We apologize for this mistake. Our measurements included: demographics (no instruments), coping style (using Self-report Coping Scale, SCS), social support (using Perceived Social Support Scale, PSSS), self-efficacy (General Self-Efficacy Scale, GSES), PTSD (using PCL-C), anxiety (using SAS), and depression (using PHQ-9). We found that we left out the description of coping style (using Self-Report Coping Scale, SCS) in the measurement section. We have added the description of the instruments employed accordingly. Besides, we adjusted the order of measurements here to match the description in the tables. Please find the changes in Page 3, from line 88 to the end of this section.

Point 5: In “Table 1. Model parameters and indexes” (on page 4) there are variables that are not described in the Measurement section. And in said table Ethnicity appears 2 times but Education does not appear.

Response 5: We noticed that some demographics and the questions on the situation related to COVID-19 have not been stated in the measurement section. We have added these and gave descriptions accordingly. Please view Page 3, from line 88 (the first paragraph of the measurement section) for changes.

About the duplicate of “Ethnicity” in Table 1, we have to apologize that we made mistakes when preparing the tables. The second “ethnicity” should be actually “educational background”. We have corrected it in Table 1.

Point 6: The Discussion section should be thoroughly reviewed and instead of presenting so many results (many of which should be moved to the Results section), the discussion should focus on the evaluation, interpretation and implications of the results obtained. Also, on page 10, in lines 187 to 189, it says “Coping style moderated the stress-emotional distress relationship, i.e., individuals who mainly adopted positive coping strategies suffered fewer symptoms of depression, compulsion-anxiety, and neurasthenia under stress, while negative coping strategies aggravated emotional distress” but in the study presented there is no evidence that analyzes have been done to find moderating variables.

Response 6: We have moved all things that should be stated under the Results section to their correct place accordingly. This included some psychometrics stated in the Methods section and some data presented in the Discussion section. Also, we re-orgaized the expression and statements in the Discussion section to make it more clear and focus on the interpretation of the results.

Point 7: Also, on page 10, in lines 187 to 189, it says “Coping style moderated the stress-emotional distress relationship, i.e., individuals who mainly adopted positive coping strategies suffered fewer symptoms of depression, compulsion-anxiety, and neurasthenia under stress, while negative coping strategies aggravated emotional distress” but in the study presented there is no evidence that analyzes have been done to find moderating variables.

Response 7: Our coping style questionnaire surveyed whether an individual's coping style was positive or negative. The higher the score, the more positive the coping style. Through model prediction, we conclude that this risk factor is one of the important factors in assessing the risk of PTSD.  On this basis, it is speculated that people with more active coping styles may be less exposed to PTSD, but the paper did not analyze its moderating effect.

Thank you again for your time spending on our manuscript. We sincerely look forward to your further opinions, and hope to have the pleasure of seeing our publication! 

With best regards,
Qu Shen, On behalf of all authors
School of Medicine, Xiamen University
12-Nov-2022

Reviewer 3 Report

Very interesting paper.

Was going to comment on cultural differences being a potential factor that would render it impossible to use this data on worldwide population but you mentioned it on the Limitations Section.

I would only suggest to avoid statements like the one where you mention the relation between the personal monthly income and the greater responsibility as it may be seen as discriminatory. Instead of stating it, you could either not mention it at all or just hypothesise. Because, is it because of greater responsibility of just because of easier access to truthful information? Or maybe just because of those people real knowledge about it (was there any relation between the income and the occupational activities?)

Author Response

It is of great pleasure for us to receive your comments, and thank you for your appreciation of the paper. We would like to thank you for your constructive comments on our manuscript.

We have modified our manuscript accordingly, and responded all comments from you below.

Point 1: English language and style: English language and style are fine/minor spell check required

Response 1: After the revision, we invited a native English-speaking colleague from Canada to improve the quality of English in the manuscript. There were a number of sentences adjusted across the text. We believe that the language now is acceptable for the review process.

Point 2: Very interesting paper. Was going to comment on cultural differences being a potential factor that would render it impossible to use this data on worldwide population but you mentioned it on the Limitations Section.

Response 2: We thank the reviewer for this comment. Regarding this point, we improved our Limitation section to make it more readable, clear, and easy to follow.

Point 3: I would only suggest to avoid statements like the one where you mention the relation between the personal monthly income and the greater responsibility as it may be seen as discriminatory. Instead of stating it, you could either not mention it at all or just hypothesise. Because, is it because of greater responsibility of just because of easier access to truthful information? Or maybe just because of those people real knowledge about it (was there any relation between the income and the occupational activities?)

Response 3: We have really realized this researcher bias and apologize for it. It is not a good statement and tends to be discriminative. We decided to delete related statements and sentences accordingly, because even a hypothesis would still be biased and without strong evidence.

You have mentioned the relationships among personal income, information accessibility, and knowledge. This is really an interesting point though out of the scope of this study. These relationships will raise potential biases when interpretating income-related outcomes. We will notice it when conducting following studies. Lastly, relationships between the income and the occupational activities should exists (according to our previous experience). We will consider more about your points on potential discrimination risk and income-related relationships in our future studies, to avoid making the same mistake. Thanks for your viewpoints.

Thank you again for your time spending on our manuscript. We sincerely look forward to your further opinions, and hope to have the pleasure of seeing our publication! 

With best regards,
Qu Shen, On behalf of all authors
School of Medicine, Xiamen University
10-Nov-2022

Round 2

Reviewer 2 Report

The manuscript “A machine learning-based risk prediction model for post-traumatic stress disorder during the COVID-19 pandemic” deals with a relevant topic. The revised manuscript has improved significantly from the initial version, so I think it can be published.